# Tibial Damage Caused by T-2 Toxin in Goslings: Bone Dysplasia, Poor Bone Quality, Hindered Chondrocyte Differentiation, and Imbalanced Bone Metabolism

**DOI:** 10.3390/ani14152281

**Published:** 2024-08-05

**Authors:** Wang Gu, Lie Hou, Qiang Bao, Qi Xu, Guohong Chen

**Affiliations:** 1College of Animal Science and Technology, Yangzhou University, Yangzhou 225009, China; dx120210144@stu.yzu.edu.cn (W.G.); hle19960225@163.com (L.H.); yan1943659975@163.com (Q.B.); 2Animal Husbandry Extension Station, Yinchuan 750001, China; 3Joint International Research Laboratory of Agriculture and Agri-Product Safety, Ministry of Education of China, Yangzhou University, Yangzhou 225009, China; 4Key Laboratory for Evaluation and Utilization of Livestock and Poultry Resources (Poultry), Ministry of Agriculture and Rural Affairs, Yangzhou University, Yangzhou 225009, China

**Keywords:** gosling, T-2 toxin, tibia, tibial growth plate, bone metabolism

## Abstract

**Simple Summary:**

Simple Summary: T-2 toxin is widely present in grain and animal feed, posing a threat to cause growth retardation and tissue damage in poultry. Goose is a sensitive waterfowl to T-2 toxin. Tibia is a representative of long bones, which is the target organ of T-2 toxin. However, tibial damage caused by T-2 toxin in geese has not been fully demonstrated. Here, Yangzhou goslings were daily gavaged with T-2 toxin at concentrations of 0, 0.5, 1.0, and 2.0 mg/kg body weight for 21 days, and the changes in bone development, bone quality, chondrocyte differentiation, and bone metabolism in the tibia were investigated. The results showed that T-2 toxin significantly inhibited tibial growth and development, accompanied by decreased bone geometry parameters and properties, hindered chondrocyte differentiation, and imbalanced bone metabolism. These findings reveal tibial damage caused by T-2 toxin in goslings, providing scientific information for understanding and alleviating the osteotoxicity of T-2 toxin in poultry production.

**Abstract:**

T-2 toxin, the most toxic type A trichothecene, is widely present in grain and animal feed, causing growth retardation and tissue damage in poultry. Geese are more sensitive to T-2 toxin than chickens and ducks. Although T-2 toxin has been reported to cause tibial growth plate (TGP) chondrodysplasia in chickens, tibial damage caused by T-2 toxin in geese has not been fully demonstrated. This study aims to investigate the adverse effects of T-2 toxin on tibial bone development, bone quality, chondrocyte differentiation, and bone metabolism. Here, forty-eight one-day-old male Yangzhou goslings were randomly divided into four groups and daily gavaged with T-2 toxin at concentrations of 0, 0.5, 1.0, and 2.0 mg/kg body weight for 21 days, respectively. The development of gosling body weight and size was determined by weighing and taking body measurements after exposure to different concentrations of T-2 toxin. Changes in tibial development and bone characteristics were determined by radiographic examination, phenotypic measurements, and bone quality and composition analyses. Chondrocyte differentiation in TGP and bone metabolism was characterized by cell morphology, tissue gene-specific expression, and serum marker levels. Results showed that T-2 toxin treatment resulted in a lower weight, volume, length, middle width, and middle circumference of the tibia in a dose-dependent manner (*p* < 0.05). Moreover, decreased bone-breaking strength, bone mineral density, and contents of ash, Ca, and P in the tibia were observed in T-2 toxin-challenged goslings (*p* < 0.05). In addition, T-2 toxin not only reduced TGP height (*p* < 0.05) but also induced TGP chondrocytes to be disorganized with reduced numbers and indistinct borders. As expected, the apoptosis-related genes (*CASP9* and *CASP3*) were significantly up-regulated in chondrocytes challenged by T-2 toxin with a dose dependence, while cell differentiation and maturation-related genes (*BMP6*, *BMP7*, *SOX9*, and *RUNX2*) were down-regulated (*p* < 0.05). Considering bone metabolism, T-2 toxin dose-dependently and significantly induced a decreased number of osteoblasts and an increased number of osteoclasts in the tibia, with inhibited patterns of osteogenesis-related genes and enzymes and increased patterns of osteoclast-related genes and enzymes (*p* < 0.05). Similarly, the serum Ca and P concentrations and parathyroid hormone, calcitonin, and 1, 25-dihydroxycholecalciferol levels decreased under T-2 toxin exposure (*p* < 0.05). In summary, 2.0 mg/kg T-2 toxin significantly inhibited tibia weight, length, width, and circumference, as well as decreased bone-breaking strength, density, and composition (ash, calcium, and phosphorus) in 21-day-old goslings compared to the control and lower dose groups. Chondrocyte differentiation in TGP was delayed by 2.0 mg/kg T-2 toxin owing to cell apoptosis. In addition, 2.0 mg/kg T-2 toxin promoted bone resorption and inhibited osteogenesis in cellular morphology, gene expression, and hormonal modulation patterns. Thus, T-2 toxin significantly inhibited tibial growth and development with a dose dependence, accompanied by decreased bone geometry parameters and properties, hindered chondrocyte differentiation, and imbalanced bone metabolism.

## 1. Introduction

T-2 toxin, a highly toxic type A trichothecene, is produced by *Fusarium* spp. and exists in animal feed [1]. Among poultry, geese are more susceptible to T-2 toxin than chickens and ducks, and goslings are particularly vulnerable [2,3]. The reduced and oxidized glutathione amount and glutathione peroxidase activity were decreased in blood and liver in geese under T-2 toxin exposure, while there was no change in chickens and ducks. T-2 toxin is easy to cause tissue damage through various pathways, including metabolism [4], inhibition of protein, DNA, and RNA synthesis [5], oxidative stress, inflammation, and cell death [6,7]. In addition to attacking organs in an active and rapid division state such as the intestines, liver, and kidneys, bone is also targeted by T-2 toxin [8,9,10]. Leg abnormalities are common manifestations of bone disorders caused by mycotoxins, raising crucial concerns in current poultry production [11].

The tibia is a key aspect of growth and development, weight support, production benefits, and animal welfare in poultry [12,13]. Tibial characteristics, including morphology, structure, composition, density, and mechanical parameters, are key indicators for evaluating tibial development and bone quality [13]. Bone strength is determined by bone mass and matrix composition, reflecting the integrity of bone quality and the ability of fracture resistance [14]. Bone strength is compromised by reducing growth, altering shape, affecting mineralization, and impairing the organic or mineral matrix composition [15]. In addition, bone mineral density (BMD) is a direct indicator of bone strength, osteoporosis, and fracture risk, determined by the mass of organics and minerals [16]. Collagen is the main component of the organic matrix, while calcium (Ca) and phosphorus (P) are the main components of the mineral matrix. Ca and P are essential elements constituting bone weight, participating in the deposition and calcification of hydroxyapatite crystals, providing stiffness and resistance to compression, and maintaining bone mass for the growth and development of bones [15,17,18]. The intake, absorption, transport, utilization, and metabolism of Ca and P, in concert with the hormone secretion of parathyroid hormone (PTH), calcitonin (CT), and 1,25-dihydroxycholecalciferol [1,25-(OH)_2_-D_3_], maintain bone growth and development and the balance of bone metabolism, ensuring the stability and health of bone tissue. However, the changes in tibial morphology, characteristics, and underlying mechanisms under T-2 exposure in poultry are unclear. Thus, elucidating the overview of tibial damage induced by T-2 toxin in goslings is instrumental in revealing the osteotoxicity of T-2 toxin in poultry.

The tibia is a representative long bone that grows primarily through endochondral osteogenesis [19]. The TGP, a layer of hyaline cartilage located between the distal epiphysis and the metaphysis of the tibia, is the ultimate target of tibial longitudinal growth [20]. Longitudinal growth of the tibia driven by TGP is intensive and rapid in young animals, whereas it stops after the fusion of TGP [19]. Tibial longitudinal growth is determined by the regulation of the proliferation, differentiation, maturation, and extracellular matrix secretion of TGP chondrocytes. The TGP is divided into a reserve zone (RZ), a proliferative zone (PZ), a hypertrophic zone (HZ), and a calcified zone (CZ) based on the different stages of chondrocyte differentiation [20,21]. The TGP is the most fragile and vulnerable part of the tibia with a limited regenerative capacity, and its damage leads to impaired bone growth [22,23]. As an environmental factor, T-2 toxin tends to cause tibial growth plate (TGP) chondrocyte damage in goslings [8,24]. In addition, despite the different durations (7, 14, and 21 days) and amounts (from 0.3 to 1.9 mg/kg) of T-2 ingestion, the tibia of chicks presented a feature similar to early tibial dyschondroplasia with defective chondrocytes and severe vascular lesions [25]. T-2 toxin inhibited the proliferation and differentiation of primary cultures of chicken TGP chondrocytes by altering cellular homeostasis [26]. T-2 toxin has been reported to cause TGP lesions in poultry, but the underlying mechanism is unclear [8,25,26]. 

The balance between osteoblast-mediated bone formation and osteoclast-mediated bone resorption is a dynamic and unifying process that reflects the state of bone metabolism [27]. Dysfunction between osteoblasts and osteoclasts can lead to abnormalities in bone morphology and structure, such as bone loss [28]. Receptor activator of nuclear factor kappa B ligand (RANKL) binding to RANK is involved in the differentiation of osteoclasts and promotes bone resorption, while osteoprotegerin (OPG) binding to RANKL can block the former binding to reduce the transcriptional activation signal of osteoclasts, thus inhibiting bone resorption [29,30]. OPG-RANKL-RANK signaling belonging to the osteoclast differentiation pathway is a key regulator of bone formation and resorption [29]. Elucidating the changes in bone metabolism under T-2 toxin exposure is fundamental to clarifying the maintenance of bone characteristics and homeostasis [31]. 

As tibial damage from T-2 toxin in poultry has been rarely reported and studies have mainly focused on TGP chondrocyte damage, the overview of tibial damage in geese under T-2 toxin exposure is unclear. Exploring tibial changes in bone characteristics, TGP, and bone metabolism under T-2 exposure provides the foundation of the structural framework and cellular mechanisms of tibial damage. Therefore, the objective of this study was to investigate the adverse effects of T-2 toxin on bone development, bone quality, chondrocyte differentiation, and bone metabolism in the tibia of Yangzhou goslings.

## 2. Materials and Methods

### 2.1. Animals and Sample Collection

The experimental procedures were approved by the Ethics Committee on Animal Experiments of Yangzhou University (Permit Number: 202201345, 9 June 2022, Government of Jiangsu Province, China). Any experimentation involving geese also adhered to the Regulations for the Administration of Affairs Concerning Experimental Animals, approved by the State Council of the People’s Republic of China. Goslings were raised in a brooding housing system at Tiange Goose Industry Co., Ltd. in Yangzhou with normal feeding management. The average temperature was controlled at about 28 °C, and the average humidity was controlled at about 65%. All goslings had free access to a basal corn-soybean meal diet (Table 1). The diet was based on the breed guidelines of the Yangzhou goose in China (GB/T 36784-2018) [32]. To reduce the difficulty in uneven stirring, individual feed intake calculation, toxin waste, and individual injury phenotype differences by T-2 toxin supplemented in feed, we fed goslings with T-2 toxin by oral gavage to explore the effect of T-2 toxin on tibial growth and development. The schematic diagram and experimental strategy for tibial damage caused by T-2 toxin in goslings are presented in Figure 1. Forty-eight one-day-old male Yangzhou goslings were randomly assigned to four groups and subjected to a 21-day oral gavage treatment with T-2 toxin at concentrations of 0, 0.5, 1.0, and 2.0 mg/kg body weight. T-2 toxin is insoluble in water but is soluble in organic solvents such as ethanol. T-2 toxin (Cat No: MSS1023-100, 100 mg, Qingdao Pribolab Biotech Co., Ltd., Qingdao, China) was dissolved in 10 mL of non-aqueous ethanol to obtain a primary solubility of 10 mg/mL of mother liquor, which was divided into ten tubes and stored at −20 °C. Before oral gavage, enzyme-free water was added to achieve the target concentration, and the daily usage was determined by daily body weight. The control received a calculated amount of ethanol-water solution.

After daily oral gavage, rearing management, and recording, the number of the remaining goslings was 12, 9, 10, and 7 in the 0, 0.5, 1.0, and 2.0 mg/kg T-2 groups, respectively. Goslings at 21 days of age were fasted overnight for 6 h before sacrifice on the 21st day. After phenotypic observation, live body weight and body size were measured, followed by blood collection. After cleaning with 75% alcohol, 2 mL of blood samples were collected from the right-wing vein, centrifuged for serum collection, and stored at −20 °C for serum biochemical tests and enzyme-linked immunosorbent assays (ELISA). Both sides of the tibia were collected after separation. The phenotypic observation and measurements of the tibia were conducted after cleaning to remove adhesions, including muscles, connective tissue, and the fibula bone. For the analysis of bone characteristics, the right tibia was collected for the tests of bone-breaking strength, BMD, and bone composition. For histological analysis, a left tibia sample (approximately 1.5 cm × 1.5 cm × 1.5 cm), including the TGP, was fixed in 10% formalin for various stainings, and tissue samples were collected and stored at −80 °C for real-time quantitative polymerase chain reaction (RT-qPCR).

### 2.2. Body Measurements

Goslings were placed on a desk for body measurements, with tape and calipers close to the body surface, keeping the movements soft. Referring to the criteria of performance terminology and measurements for poultry in China (NY/T 823-2020) [33], body size indexes of goslings were measured with tape and calipers on the 21st day, including body slope length, keel length, chest depth, chest width, shank length, shank circumference, pelvis width, and half-diving depth. Body slope length, measured by tape, is the distance between the shoulder joint and ipsilateral ischial tuberosity. Keel length, measured by tape, is the distance from the front end to the end of the keel. Chest depth, measured by calipers, is the distance from the first thoracic vertebra to the anterior edge of the keel. Chest width, measured by calipers, is the distance between two shoulder joints. Shank length, measured by calipers, is the linear distance from the superior metatarsal joint to the third and fourth toes. Shank circumference, measured by tape, is the circumference length of the middle shank. Pelvis width, measured by calipers, is the distance between two hip bone nodules. Half-diving depth, measured by tape, is the distance from the beak tip of the waterfowl to the midpoint of the connection between the two hip bone nodules. 

### 2.3. Tibial Phenotypic Measurements and Imaging Analysis

After removing the tarsal bone and adhesions, the weight, apparent volume, length, middle width, and middle circumference of the tibia were measured, followed by the phenotypic image capture, X-ray examination, and CT scan [34,35,36]. For the X-ray examination (YEMA clearXvet DR50; YEMA ELECTRONICS, LLC, Salt Lake, UT, USA) and CT scan (InsitumCT 768; SinoVision Technology (Beijing) Co., Ltd., Beijing, China)), the tibia samples were placed at the target position for detection with default parameters (X-ray: 6mAs 50 KV; CT: layer thickness 0.625 mm 100 KV), and then the images were analyzed by the physician. 

### 2.4. Tibial Bone-Breaking Strength Determination

We placed the tibia on the holder, started the program, and recorded the values after the tibia was sheared and fractured. The bone-breaking strength of the middle position of the tibia was measured by a quasi-static three-point bending test using a texture analyzer (TMS-PRO; Food Technology Co., Sterling, VA, USA) [34,37]. The system automatically found the zero displacement point, and the probe rose back to the sample’s surface with a return distance of 20 mm and a return speed of 60 mm/min. The bone-breaking strength was determined by identifying the failure point (peak) of each loading curve.

### 2.5. Tibial Bone Mineral Density Determination

After measuring bone-breaking strength, the breaking volume (cm^3^) was measured based on the change in the water amount (mL) in the measuring cylinder. Then, the tibia was dried at 105 °C for 24 h to weigh the dewatered dry weight (g). Subsequently, the tibia was degreased by petroleum ether extraction for 7 d, dried at 105 °C for 24 h, and the defatted dry weight (g) was measured. At last, the tibial density (g/cm^3^) was calculated as the ratio of fat-free weight to breaking volume [35]. 

### 2.6. Tibial Bone Composition Determination

To determine the content of inorganic components, the defatted tibia was crushed, heat-carbonized, and calcined in a muffle furnace at 550 °C for 24 h. Then, the ash content was determined after cooling in the dryer at room temperature [17]. For Ca and P determination, the sample was weighed and put into the crucible for carbonizing. Next, 10 mL of hydrochloric acid solution and several drops of concentrated nitric acid were added to the crucible, boiled, cooled, and transferred to a 100 mL volumetric flask. The sample decomposition liquid was obtained in constant volume with water. Finally, Ca was determined by the EDTA method, and P was determined by molybdenum yellow colorimetry [17].

### 2.7. Tibial Stainings by Hematoxylin–Eosin (HE), Alcian Blue (AB), Safranin O-Fast Green (SF), and Tartrate-Resistant Acid Phosphatase (TRAP)

Tibial stainings were conducted according to the guidelines of the HE, AB, SF, and TRAP dye solution kit sets (CAT NO: G1076, G1027, G1053, and G1050, Wuhan Servicebio Biotechnology Co., Ltd., Wuhan, China) [38,39]. In brief, the tibia tissue fixed in 10% formalin was decalcified with 0.5 M EDTA for 1 month. Subsequently, the sample was dehydrated until transparent and embedded in paraffin. Then, the section was prepared, placed in water for extension, and dried. Next, the section was placed in xylene, ethanol, and 75% ethanol and washed with water. For HE staining, the section was stained with hematoxylin, followed by staining with eosin. For AB staining, the section was stained with alcian blue dye for 10~15 min, followed by impregnating nuclear solid red dye for 3 min. For SF staining, the section was stained in a fast green dye solution for 1~5 min, washed away the excess dye solution until the cartilage was colorless, and soaked in 1% hydrochloric acid and alcohol for 10 s. After that, the section was stained in a saffron dye solution for 1~5 s, put into ethanol for three rapid dehydrations of 5 s, 2 s, and 10 s, respectively, and kept in the fourth cylinder. For TRAP staining, the section was incubated with distilled water at 37 °C for 2 h in the wet box, followed by adding the filtered and prepared Trap incubation solution and putting it in the oven at 37 °C for 20 min. Finally, the section was regularly dehydrated and sealed with neutral gum. The relative histological images for further observation, measurement, and cell counts were captured using a micro-image analysis system (CaseViewer 2.4, 3DHISTECH Ltd., Budapest, Hungary).

### 2.8. RT-qPCR

Total RNA extraction, reverse transcription, and RT-qPCR were conducted under the guidance of (R401-01, R212-01, Q111-02, Nanjing Vazyme Biotech Co., Ltd., Nanjing, China). The primers were designed by mRNA reference sequences in GenBank (Table 2), and the actin beta (ACTB) gene was used as an internal reference gene. The data from RT-qPCR were analyzed with the 2^−∆∆Ct^ method using the Ct value.

### 2.9. Serum Biochemical Tests

Serum samples were sent to the Yangzhou University Animal Hospital for the determination of Ca (CA01, Ningbo Purebio Biotechnology Co., Ltd., Ningbo, China) and P (PHO01, Ningbo Purebio Biotechnology Co., Ltd., Ningbo, China) concentrations using a fully automatic biochemical analyzer (AU480, Beckman Coulter K.K., Tokyo, Japan) [8]. 

### 2.10. ELISA

Serum alkaline phosphatase (ALP; A059-1-1), TRAP (A058-1-1), osteocalcin (OCN/BGP; MM-3419402), PTH (MM-215102), CT (MM-3421202), and 1,25-(OH)_2_-D_3_ (D751006-0048) levels were detected using ELISA following the guidance from commercial kits (ALP, TRAP: Nanjing Jiancheng Bioengineering Institute., Nanjing, China; BGP, PTH, CT: Jiangsu Meimian Industrial Co., Ltd., Nanjing, China; 1,25-(OH)_2_-D_3_: Sangon Biotech (Shanghai) Co., Ltd., Shanghai, China). 

### 2.11. Statistical Analysis

The statistical significance of differences among different T-2 toxin-contaminated groups was analyzed by SPSS software (version 25.0; IBM Corp., Armonk, NY, USA). Based on the Shapiro–Wilk (*n* < 50) and homogeneity tests to ensure the normality and homogeneity of variances, the data were expressed as the mean ± standard error and subjected to a one-way ANOVA followed by a Scheffe multiple range test (*n* = 12, 9, 10, 7) or Tukey’s honest significant difference multiple range test (*n* = 6 in RT-qPCR). Statistical significance was set at *p* < 0.05.

## 3. Results

### 3.1. Effects of T-2 Toxin on Body Weight and Body Size

Body weight and body size are basic criteria for evaluating appearance, performance, and bone growth and development. T-2 toxin dose-dependently and significantly reduced body weight and body size indexes in Yangzhou goslings, including body slope length, keel length, chest depth, shank length, shank circumference, pelvis width, and half-diving depth (*p* < 0.05; Table 3). Thus, T-2 toxin has adverse effects on the growth and development of bones in goslings, suggesting bone abnormalities.

### 3.2. Tibial Phenotype, Quality, and Composition under T-2 Toxin Exposure

The tibia serves as a model long bone, which is representative of the overall bone system in poultry studies. In terms of bone morphology, T-2 toxin dose-dependently and significantly reduced the growth and development of the tibia, especially in the changes in fresh weight, volume, length, middle width, and middle circumference (*p* < 0.05; Table 4). In addition to bone geometry, bone-breaking strength, BMD, and bone matrix composition are common predictors of bone development and quality. T-2 toxin dose-dependently and significantly reduced the tibial bone-breaking strength, BMD, and inorganic matrix components, including the content of ash, Ca, and P (*p* < 0.05; Table 4). Consistent with the analysis of size, structure, and biomechanical properties, phenotypic and radiographic observations also revealed T-2 toxin-induced tibial abnormalities (Figure 2A–C). In X-ray examination, the TGP exists as a dark line without fracture in all T-2 toxin groups (Figure 2B). As the concentration of T-2 toxin increased, the shadowed areas on both sides of the tibia became gradually larger, and the bone density decreased sequentially in the CT scan (Figure 2C).

### 3.3. Tibial Growth Plate Lesions Induced by T-2 Toxin

TGP is key to the longitudinal growth of the long bones. Compared with the control, T-2 toxin significantly reduced the apparent height of TGP in a dose-dependent manner (*p* < 0.05; Figure 3A). Consistent with the anatomic observation, TGP chondrocytes underwent abnormal changes through various stains (Figure 3B). HE staining is advantageous in observing the hierarchical structure, cyclic morphological changes, and damage of chondrocytes in TGP, but the microstructure and extracellular matrix components of chondrocytes are vague and cannot be measured and distinguished. However, AB staining and SF staining are easier to show cartilage and bone tissues of different maturities than HE staining with different colors. Compared with the control, the chondrocytes in the RZ and PZ of various T-2 toxin groups showed growth arrest, were deformed, and were smaller after being squeezed, and the cell arrangement was disordered. Moreover, chondrocyte filling was reduced in the HZ, more vacuoles, excessive apoptosis, and necrosis were observed, and calcification was reduced in the CZ (Figure 3B; Appendix A). In addition, gene expressions of bone morphogenetic protein 6 (*BMP6*), *BMP7*, SRY-box transcription factor 9 (*SOX9*), and RUNX family transcription factor 2 (*RUNX2*) were significantly decreased by T-2 toxin in a dose-dependent manner in TGP (*p* < 0.05; Figure 3C). Tissue quantitative results also showed that the expression of the antiapoptotic gene of BCL2 apoptosis regulator (*BCL2*) was significantly decreased, and expressions of the pro-apoptotic genes of caspase 9 (*CASP9*) and *CASP3* were significantly increased by T-2 toxin, exhibiting a dose effect (*p* < 0.05; Figure 3D).

### 3.4. Imbalance of Osteogenesis and Osteoclastogenesis Caused by T-2 Toxin

Bone metabolism is controlled by two opposing and united processes: bone formation by osteoblasts (OB) and bone resorption by osteoclasts (OC). Compared with the control, T-2 toxin dose-dependently induced a decrease in the number of osteoblasts and an increase in the number of osteoclasts in the tibia (*p* < 0.05; Figure 4A; Appendix A). It was also found that gene expressions of *BGP*, *OPG*, and *OPG*/*RANKL* ratio were decreased by T-2 toxin in a dose-dependent manner in the tibia, while gene expressions of *RANKL* and *RANK* were increased (*p* < 0.05; Figure 4B). Consistent with the results of cell morphology observation and tissue quantification, T-2 toxin caused a significant increase in the enzyme activities of serum ALP and TRAP and a significant decrease in BGP activity (*p* < 0.05; Figure 4C). In addition, T-2 toxin dose-dependently and significantly reduced serum calcium concentration compared with the control, but the phosphorus concentration was significantly decreased only in the 2.0 mg/kg T-2 toxin group. Considering the hormonal regulation of calcium-phosphorus metabolism and bone metabolism, T-2 toxin dose-dependently and significantly reduced serum PTH, CT, and 1,25-(OH)_2_-D_3_ levels (*p* < 0.05; Figure 4D). 

## 4. Discussion

T-2 toxin is a type A trichothecene produced by *Fusarium* species and is widespread in corn and grain feed, posing a threat to growth and development and animal welfare in poultry [40,41]. Goose is a representative of waterfowl animals sensitive to T-2 toxin exposure [3]. There is less data on tissue damage caused by T-2 toxin in geese than in chickens and ducks. Due to the complexity of physicochemical properties and damage mechanisms, there are no effective strategies to prevent and mitigate T-2 toxin contamination or poisoning [40,42]. Food refusal and growth retardation, which inhibit increases in body weight and body size, are intuitive phenomena of T-2 toxin poisoning in animals [8,43]. T-2 toxin causes contact irritation and metabolic toxic damage to the rostrum, oral cavity, and gastrointestinal tract, leading to food refusal and growth retardation [8,44]. Under T-2 exposure, the decline in food intake is regulated by various appetitive central and peripheral modulators, gut satiety hormones, proinflammatory cytokines, and growth hormone deficiency [8,45,46,47,48]. Moreover, T-2 toxin inhibits the synthesis and secretion of key molecular precursors for nutritional and immune needs by blocking protein, DNA, and RNA synthesis [8,49]. In addition, T-2 toxin induces tissue damage through oxidative stress, inflammation, apoptosis, and autophagy [8,50,51]. This results in a malnourished and stunted physiological environment for body growth and development, especially body weight loss. Body weight and body size are also basic indicators for evaluating the appearance, growth and development, production performance, and health status of poultry, and there is a certain degree of synchronization and correlation between them [52,53]. In this study, the high dose of 2.0 mg/kg of T-2 toxin had a significant inhibitory effect on body weight gain and body size development compared to control and low doses. This finding serves as a valid marker of T-2 poisoning in geese, suggesting bone problems. Leg abnormalities, common and external manifestations of bone disorders, are major threats to global poultry production, raising welfare and economic concerns. The tibia, a rapidly growing and dividing organ, is a target of T-2 toxin exposure [8]. However, studies on tibial damage induced by T-2 toxin in poultry are limited and mainly focused on TGP chondrocyte damage in chickens, while the overview of tibial damage in geese under T-2 toxin exposure is unclear. Therefore, clarification of the characteristics and intrinsic mechanisms of tibial damage caused by T-2 toxin in goslings is beneficial for understanding and alleviating the osteotoxicity of T-2 toxin in poultry production.

The tibia is representative of long bones that form the basic scaffolding of the body, support body weight, and maintain growth and development, productive performance, and animal welfare in poultry [13]. Morphology, physical and chemical properties, microstructure, bone formation, metabolism, and remodeling processes are essential aspects affecting tibial growth, development, and health [54,55]. In this study, T-2 toxin significantly inhibited tibial growth and development in a dose-dependent manner, mainly including significant decreases in weight, volume, length, middle width, and middle circumference by morphological observations and measurements. In addition, bone strength, BMD, and bone composition are important parameters for evaluating bone characteristics and quality. Here, we found that T-2 toxin significantly reduced tibial bone-breaking strength and BMD with a dose dependence, which was consistent with the CT scan. In terms of bone composition, the contents of ash, calcium, and phosphorus were also significantly reduced under T-2 toxin exposure in a dose-dependent manner. As an antinutritional environmental factor, T-2 toxin can inhibit bone growth, cause bone fragility, and decrease bone strength by affecting the metabolism of multifactors such as minerals and vitamins [15]. T-2 toxin reduces ingestion and utilization of the minerals calcium and phosphorus by inhibiting feed intake and intestinal absorption. Inadequate supply of calcium and phosphorus leads to poor bone growth and development, impaired bone structure, poor bone formation and mineralization, and imbalanced bone health. Calcium and phosphorus are the basic mineral elements that shape bones and are deposited in bone tissue in the form of hydroxyapatite, which maintains bone strength and resistance to pressure [56]. Decreases in bone strength and density, as well as abnormal changes in the bone matrix, lead to the development of bone diseases such as spontaneous fractures, osteoporosis, and dysplasia [14,28,57]. Thus, T-2 toxin significantly causes retardation of tibial growth and development with poor bone quality in a dose-dependent manner, which is unfavorable for higher growth rates in modern meat geese production.

The TGP is recognized as a determinant of tibial longitudinal growth in young animals [19]. In this study, the TGP was observed to be a dark line in X-ray examination in goslings, and the TGP height was reduced by T-2 toxin with a dose dependence, which is a symbol of tibia abnormalities. The tibia is formed by endochondral bone formation, where persistent growth plates progress from proliferation, maturation, and hypertrophy of growth plate chondrocytes to mineralization of the cartilaginous matrix to form an osseous tissue [19]. Here, we found that the chondrocytes in TGP were disorganized, indistinctly bordered, and sparse in number, accompanied by more vacuoles, excessive apoptosis, necrosis, and poor calcification. The development of TGP chondrocytes is regulated by secretory and transcriptional factors. BMPs are the secretory proteins produced by chondrocytes to regulate the morphological heterogeneity of TGP chondrocytes and proper development [58,59]. *BMP7* is expressed in proliferative chondrocytes, whereas *BMP6* is expressed in pre-hypertrophic and hypertrophic chondrocytes, thereby increasing the number of proliferative chondrocytes and the thickness of the proliferative zone [60,61,62]. *SOX9* is an important transcription factor for chondrocyte proliferation and delays the onset of chondrocyte hypertrophy, while *RUNX2* is an essential transcription factor that drives differentiation and hypertrophy of proliferative chondrocytes [19,63,64]. In the present study, the expressions of *BMP6*, *BMP7*, *SOX9*, and *RUNX2* were decreased by T-2 toxin, suggesting a decline in proliferation and hypertrophy of chondrocytes. In addition, T-2 toxin has been reported to induce apoptosis in TGP chondrocytes, thereby causing thinning of the growth plate and susceptibility to fracture [65,66]. Apoptosis is the programmed death of cells under certain conditions to maintain a steady number of cells with normal function and a stable internal environment [67]. However, excessive apoptosis disrupts the dynamic balance of cell numbers and function. *BCL2* is a key antiapoptotic gene, while *CASPs* are pro-apoptotic genes [68]. Here, we also found that *BCL2* gene expression was significantly decreased, and gene expressions of *CASP9* and *CASP3* were significantly increased by T-2 toxin with a dose effect, consistent with changes in cell morphology. However, there was no significant change in the gene expression of *BAK1*, suggesting that other pro-apoptotic genes may be involved. Clarifying the expression changes in more apoptosis-related genes could help to better explore the osteotoxicity of T-2 toxin. In future studies, RNA-seq will facilitate the identification and extensive investigation of key genes related to apoptosis. Hence, T-2 toxin significantly induces impaired proliferation, differentiation, and maturation of TGP chondrocytes with abnormal cell morphology, down-regulated expression of chondrogenesis-related genes, and excessive apoptosis. These TGP chondrocyte damages ultimately lead to a dose-dependent reduction in TGP height and poor longitudinal growth. 

The tibia is a metabolically active organ that undergoes continuous updating and remodeling through bone metabolism to maintain bone characteristics and homeostasis [69]. Bone morphogenesis and reconstruction are controlled by two opposing and unified processes, where the bone formation process is mediated by OB and the bone resorption process is mediated by OC [27,69]. Here, T-2 toxin significantly induced an increase in the number of osteoclasts and a decrease in the number of osteoblasts in the tibia of goslings with a dose effect, suggesting an imbalance in bone metabolism. Imbalances in bone metabolism of bone formation and resorption lead to bone loss and abnormal structure, which further increases bone fragility and fracture risk, causing osteoporosis [69]. From the view of molecular mechanisms of bone metabolism, OPG-RANKL-RANK signaling is the key factor in determining the dominant orientation of osteogenesis or osteoclastogenesis, thus regulating the balance of bone formation and resorption [29]. Both RANKL produced by osteoblasts and RANK produced by osteoclasts promote osteoclast differentiation through cell-dependent contacts [29,69]. OPG, a soluble protein produced and secreted mainly by osteoblasts, competitively binds to RANKL, inhibits osteoclast differentiation, and promotes osteoclast apoptosis [30,69]. The OPG/RANKL ratio is an essential indicator of tibia development and integrity due to its effect on the biological effectiveness of RANKL [70]. We discovered decreased patterns of *BGP* and *OPG* gene expressions as well as the *OPG*/*RANKL* ratio under T-2 exposure with a dose effect while increasing patterns of *RANKL* and *RANL*. These significant changes indicated that the transcription level of osteoblasts was decreased while the transcription level of osteoclasts was increased, consistent with changes in cell numbers. ALP and TRAP are important phosphatases in bone remodeling and are responsible for bone formation and resorption, respectively [71]. In this study, the serum ALP and TRAP enzyme activities were increased, and BGP enzyme activity was decreased by T-2 toxin in a dose-dependent manner, consistent with the decrease in the number of osteoblasts and the increase in the number of osteoclasts through specific stainings. The increase in serum alkaline phosphatase is influenced by liver damage under T-2 exposure and is not representative of increased osteogenesis [8,72]. BGP was secreted by osteoblasts to maintain osteoblast differentiation and the normal rate of bone mineralization, one symbol of bone formation [73]. During bone growth and development, blood Ca and P concentrations are dynamically updated to maintain a balance of bone metabolism. Ca and P are ingested through the ration, absorbed, transported, utilized, and metabolized through the intestinal tract, and maintained at a relatively stable circulating level in the blood. However, T-2 toxin has been reported to reduce the apparent metabolic rate of Ca and total P, leading to an insufficient supply [74]. Deficiencies or excesses of Ca and P, or imbalances in their ratios, interfere with the absorption and deposition of Ca and P, thus impairing bone growth and development [75,76,77]. Compared with the control, the contents of Ca and P in the serum and tibia of T-2 toxin-exposed goslings decreased in a dose-dependent manner. Ca deficiency leads to poor bone mineralization; affects bone mineral density, bone strength, and bone ash content; and promotes the development of osteoporosis [78]. Moreover, there is a bidirectional effect between Ca and P concentrations and the secretion of hormones such as PTH, CT, and 1,25-(OH)_2_-D_3_, thereby co-regulating bone metabolism processes of bone formation and resorption to maintain bone homeostasis [28]. In addition to elevating blood Ca, PTH has the dual effect of promoting bone resorption at high doses and bone formation at low doses [79,80]. PTH is osteogenic by acting directly on mesenchymal stem cells and osteoblast lineages, while it regulates bone resorption by acting indirectly on osteoclasts and osteoclast precursors [80]. PTH affects the activity of osteoclasts through the OPG/RANKL system, and the ratio of OPG to RANKL determines the differentiation and function of osteoclasts [79]. CT reduces blood Ca and phosphorus by inhibiting the formation of osteoclasts and enhancing the osteogenic process in a positive feedback manner [81]. Moreover, 1,25-(OH)_2_-D_3_, a highly active form of vitamin D_3_ in the body, promotes the rise of blood Ca by a net effect of osteolysis [82,83]. Here, we found that T-2 toxin significantly reduced serum PTH, CT, and 1,25-(OH)_2_-D_3_ hormone levels in a dose-dependent manner, consistent with the lower Ca and P concentrations, reduced osteogenesis, and increased osteolysis. Therefore, T-2 toxin significantly enhanced bone resorption but weakened bone formation with consistent cell morphology, gene expressions, serum enzyme activities, serum Ca and P concentrations, and serum hormone levels, resulting in imbalanced bone metabolism.

## 5. Conclusions

In conclusion, 2.0 mg/kg T-2 toxin significantly inhibited tibia weight, length, width, and circumference, as well as decreased bone-breaking strength, density, and composition (ash, calcium, and phosphorus) in 21-day-old goslings compared to the control and lower dose groups. Chondrocyte differentiation in TGP was delayed by 2.0 mg/kg T-2 toxin owing to cell apoptosis. In addition, 2.0 mg/kg T-2 toxin promoted bone resorption and inhibited osteogenesis in cellular morphology, gene expression, and hormonal modulation patterns. Thus, T-2 toxin significantly inhibits tibial growth and development in a dose-dependent manner, manifested in decreased bone geometry parameters and properties, hindered chondrocyte differentiation, and imbalanced bone metabolism. These findings reveal tibia damage caused by T-2 toxin in goslings, providing scientific information for understanding and alleviating the osteotoxicity of T-2 toxin in poultry production.

## Figures and Tables

**Figure 1 animals-14-02281-f001:**
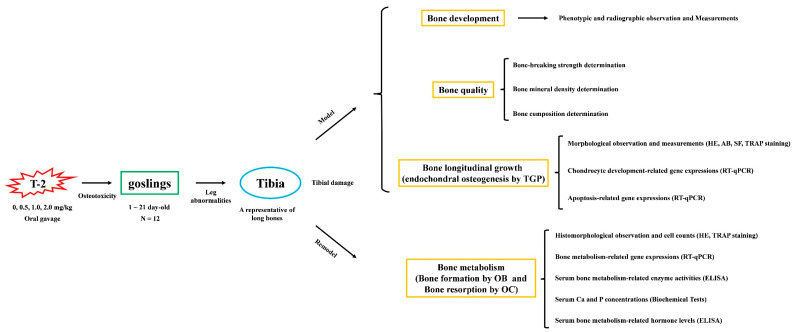
The schematic diagram and experimental strategy of tibial damage caused by T-2 toxin in goslings. T-2, T-2 toxin; TGP, tibial growth plate; HE, hematoxylin–eosin; AB, alcian blue; SF, safranin O-fast green; TRAP, tartrate-resistant acid phosphatase; RT-qPCR, real-time quantitative PCR; OB, osteoblast; OC, osteoclast.

**Figure 2 animals-14-02281-f002:**
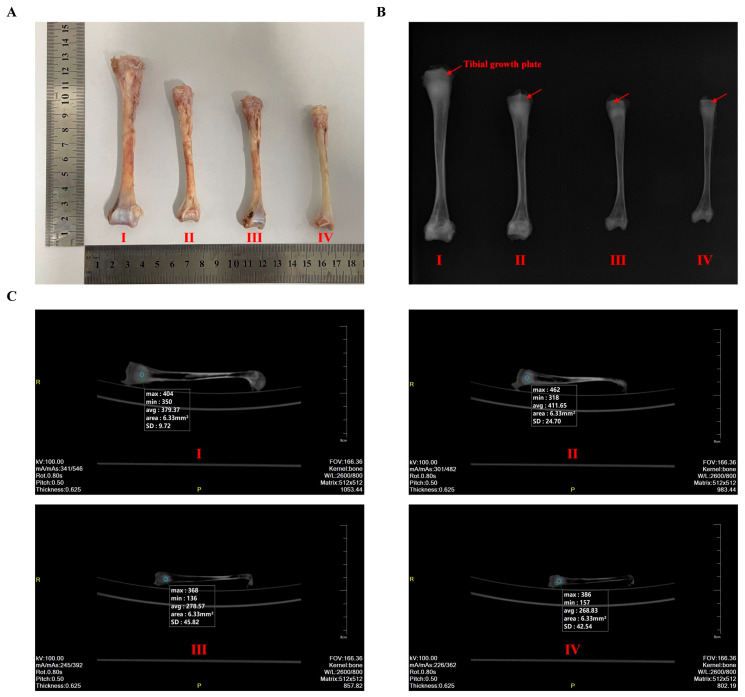
Phenotypic and radiographic observations of the tibia in goslings under T-2 toxin exposure through morphological image capture (**A**), X-ray examination (**B**), and CT scan (**C**). Ⅰ, 0 mg/kg T-2 toxin group; Ⅱ, 0.5 mg/kg T-2 toxin group; Ⅲ, 1.0 mg/kg T-2 toxin group; and Ⅳ, 2.0 mg/kg T-2 toxin group. The red arrow refers to the tibial growth plate. CT values in the selected area were detected, indicating bone mineral density in the bone plane.

**Figure 3 animals-14-02281-f003:**
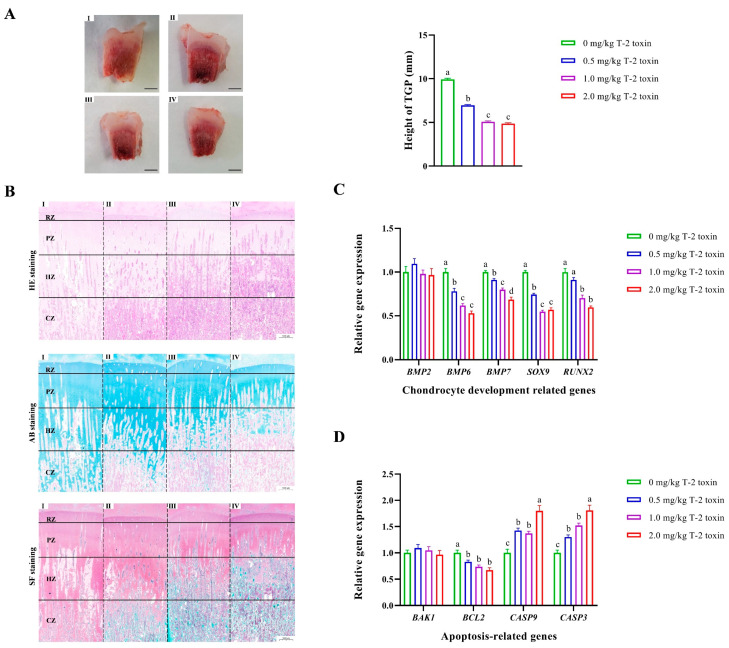
Morphological and structural changes in chondrocytes in the tibial growth plate (TGP) under T-2 toxin exposure. Ⅰ, 0 mg/kg T-2 toxin group; Ⅱ, 0.5 mg/kg T-2 toxin group; Ⅲ, 1.0 mg/kg T-2 toxin group; and Ⅳ, 2.0 mg/kg T-2 toxin group; (**A**) The phenotype of the tibial growth plate with a scale bar of 10 mm. Differences in TGP height were analyzed by one-way ANOVA analysis. *n* = 12, 9, 10, and 7, respectively. ^a–c^ Means within a row with no common superscript differ significantly (*p* < 0.05). (**B**) Chondrocyte damage in the reserve zone (RZ), proliferative zone (PZ), hypertrophic zone (HZ), and calcified zone (CZ) of TGP through hematoxylin–eosin (HE), alcian blue (AB), and safranin O-fast green (SF) staining with a scale bar of 1000 μm. The representative images of higher magnifications of TGP with a scale bar of 20 μm in HE staining are shown in Appendix A. (**C**) Expression patterns of chondrocyte development-related genes in TGP chondrocytes. *BMP*, bone morphogenetic protein; *SOX9*, SRY-box transcription factor 9; *RUNX2*, RUNX family transcription factor 2. Differences in gene expressions were analyzed by one-way ANOVA analysis. *n* = 6. ^a–d^ Means within a row with no common superscript differ significantly (*p* < 0.05). (**D**) Expression patterns of apoptosis-related genes in TGP chondrocytes. *BAK1*, BCL2 antagonist/killer 1; *BCL2*, BCL2 apoptosis regulator; *CASP*, caspase. Differences in gene expressions were analyzed by one-way ANOVA analysis. *n* = 6. ^a–c^ Means within a row with no common superscript differ significantly (*p* < 0.05).

**Figure 4 animals-14-02281-f004:**
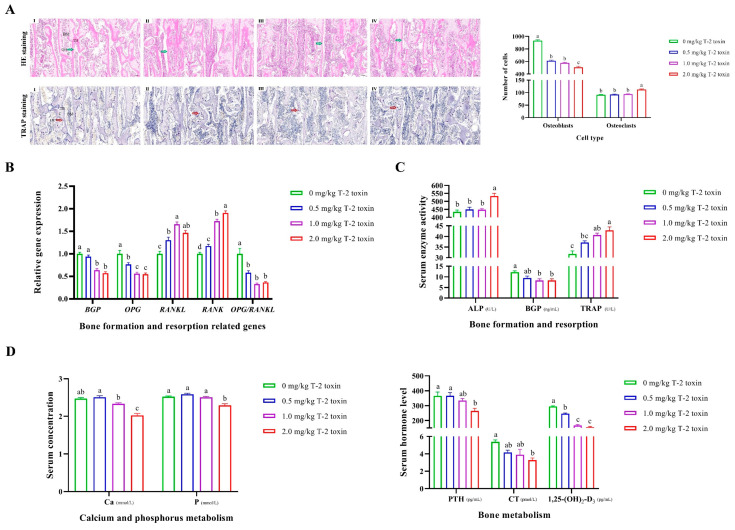
Effects of T-2 toxin on bone metabolism: bone formation (osteogenesis) and resorption (osteoclastogenesis) in the calcified zone of the tibia. (**A**) Changes in the number of osteoblasts and osteoclasts with a scale bar of 100 μm. Ⅰ, 0 mg/kg T-2 toxin group; Ⅱ, 0.5 mg/kg T-2 toxin group; Ⅲ, 1.0 mg/kg T-2 toxin group; and Ⅳ, 2.0 mg/kg T-2 toxin group; HE, hematoxylin–eosin; TRAP, tartrate-resistant acid phosphatase; OB, osteoblast (green arrow); OC, osteoclast (red arrow); C, chondrocyte; TB, trabecular bone; BM, bone marrow. Osteoblasts are polygonal or cuboidal in shape, arranged in a single layer on the surface of the newly formed bone matrix. Osteoclasts are large multinucleated cells (*n* ≥ 3) that usually show intense red or purple coloring in TRAP staining. Osteoblasts and osteoclasts were counted in five randomly selected fields of HE and TRAP staining, respectively, with a scale bar of 100 μm, and the mean values were used as a representative value for the individual sample within the group for subsequent significance analysis. Differences in cell numbers were analyzed by one-way ANOVA analysis. *n* = 12, 9, 10, and 7, respectively. ^a–c^ Means within a row with no common superscript differ significantly (*p* < 0.05). The representative images of higher magnifications of osteoblasts and osteoclasts with a scale bar of 20 μm are shown in Appendix A. (**B**) Changes in the expression of genes associated with bone formation and bone resorption in the tibia. *BGP*, bone gamma-carboxyglutamate protein; *OPG*, osteoprotegerin; *RANKL*, receptor activator of nuclear factor kappa B ligand; *RANK*, receptor activator of nuclear factor kappa B. Differences in gene expressions were analyzed by one-way ANOVA analysis. *n* = 6. ^a–d^ Means within a row with no common superscript differ significantly (*p* < 0.05). (**C**) Changes in the serum activity of enzymes associated with bone formation and bone resorption. ALP, alkaline phosphatase. Differences in enzyme activities were analyzed by one-way ANOVA analysis. *n* = 12, 9, 10, and 7, respectively. ^a–c^ Means within a row with no common superscript differ significantly (*p* < 0.05). (**D**) Changes in the serum calcium (Ca) and phosphorus (P) concentrations and hormone levels associated with bone metabolism. PTH, parathyroid hormone; CT, calcitonin; 1,25-(OH)_2_-D_3_, 1,25-dihydroxyvitamin D_3_. Differences in Ca and P concentrations and enzyme activities were analyzed by one-way ANOVA analysis. *n* = 12, 9, 10, and 7, respectively. ^a–c^ Means within a row with no common superscript differ significantly (*p* < 0.05).

**Table 1 animals-14-02281-t001:** Ingredient composition and nutritional level of the diet for goslings.

Ingredient Composition (%)		Nutritional Level (%)	
Corn	66.60	DM	87.49
Soybean meal	26.50	ME (MJ/kg)	11.64
Rice husk	0.50	Crude protein	17.04
Stone powder	0.50	Crude fiber	3.66
Calcium hydrogen phosphate	1.25	Crude fat	2.48
Methionine	0.35	Ca	0.68
Salt	0.50	Total phosphorus	0.66
Premix	3.80	Available phosphorus	0.41
Total	100.00	Lysine	0.86
		Methionine	0.60

Note: The per kilogram premix contains 1,200,000 IU VA, 400,000 IU VD, 1800 IU VE, 150 mg VK, 60 mg VB_1_, 600 mg VB_2_, 200 mg VB_6_, 1 mg VB_12_, 3 g niacin, 900 mg D-pantothenic acid, 50 mg folic acid, 4 mg biotin, 35 g choline, 6 g Fe, 1 g Cu, 9.5 g Mn, 9 g Zn, 30 mg Se, and 50 mg I. Nutrient levels were calculated as values.

**Table 2 animals-14-02281-t002:** Information on gene primers for RT-qPCR.

Gene	Accession Number	Sequence (5′→3′)	Product Length (bp)
*BMP2*	XM_048060962.1	F: CAAACAGCGTAAACGCCACA	131
		R: GACATTCCCCGTGGCAGTAA	
*BMP6*	XM_048072067.1	F: GCCTCCTCGGGCTTCCTCTA	293
		R: CTCATGACCATGTCAGCGTCG	
*BMP7*	XM_048072573.1	F: TTGTTCCTGCTCGACTCTCG	124
		R: CAGATAGCTGCAGGCCAAGA	
*SOX9*	XM_048064463.1	F: GCAGCTCACCAGACCCTAAA	127
		R: GCAGGAAAAGTCTGCGGAAG	
*RUNX2*	XM_048065734.1	F: TGCCACTTCACCACCAACTT	139
		R: AGGCGGTTTGGGATGTAAGG	
*BAK1*	XM_013197006.2	F: CAGCCCACCAAGGAGAA	153
		R: GAGGAAGCCCGTTATGC	
*BCL2*	XM_048076100.1	F: ATGACCGAGTACCTGAACCG	155
		R: GCTCCCACCAGAACCAAAC	
*CASP9*	XM_048067306.1	F: TTCCAGGCTCTGTCGGGTAA	150
		R: GTCCAGCGTTTCCACATACCA	
*CASP3*	XM_048078363.1	F: CTGGTATTGAGGCAGACAGTGG	158
		R: CAGCACCCTACACAGAGACTGAA	
*BGP*	XM_048054300.1	F: TTGGGGTTTTAAGAGGTGCTGG	231
		R: GCAGACACGCTAGGAGCATT	
*OPG*	XM_013185061.2	F: CATCTCAACACACTGATGGCAAG	147
		R: GATGGTGTCTTGGTCTCCATTCT	
*RANKL*	XM_013179680	F: ACCTGACTAAAAGAGGGCTTCAG	102
		R: AGTATTTGGTGCTTCCTCCCTTC	
*RANK*	XM_048076117.1	F: CAGAGATGCGTTCGTTGCTG	230
		R: CAGGTGGGAAATGGTCGTGA	
*ACTB*	XM_013174886.1	F: GCACCCAGCACGATGAAAAT	150
		R: GACAATGGAGGGTCCGGATT	

Gene abbreviations: BMP, bone morphogenetic protein; SOX9, SRY-box transcription factor 9; RUNX2, RUNX family transcription factor 2; BAK1, BCL2 antagonist/killer 1; BCL2, BCL2 apoptosis regulator; CASP, caspase; BGP, bone gamma-carboxyglutamate (gla) protein (OCN, osteocalcin); OPG, osteoprotegerin (TNFRSF11B, TNF receptor superfamily member 11b); RANKL, receptor activator of nuclear factor kappa B ligand (OPGL, osteoprotegerin ligand; TNFSF11, TNF superfamily member 11); RANK, receptor activator of nuclear factor kappa B (TNFRSF11A, TNF receptor superfamily member 11a); ACTB, actin beta.

**Table 3 animals-14-02281-t003:** Effects of T-2 toxin on body weight and body size of Yangzhou goslings.

Item	0 mg/kg	0.5 mg/kg	1.0 mg/kg	2.0 mg/kg	*p*-Value
Body weight (g)	646.35 ± 30.84 ^a^	576.46 ± 21.61 ^a^	553.24 ± 33.77 ^a^	413.30 ± 31.98 ^b^	0.000
Body slope length (cm)	13.98 ± 0.16 ^a^	12.67 ± 0.37 ^ab^	12.47 ± 0.38 ^b^	11.01 ± 0.39 ^c^	0.000
Keel length (cm)	6.73 ± 0.22 ^a^	5.49 ± 0.09 ^b^	5.44 ± 0.13 ^b^	5.31 ± 0.18 ^b^	0.000
Chest depth (cm)	5.07 ± 0.13 ^a^	5.18 ± 0.15 ^a^	4.82 ± 0.10 ^a^	4.11 ± 0.07 ^b^	0.000
Chest width (cm)	5.39 ± 0.09	5.30 ± 0.16	5.03 ± 0.14	4.84 ± 0.16	0.029
Shank length (mm)	67.78 ± 1.20 ^a^	56.71 ± 0.73 ^b^	55.00 ± 0.60 ^bc^	50.63 ± 1.63 ^c^	0.000
Shank circumference (cm)	3.74 ± 0.09 ^a^	3.53 ± 0.10 ^ab^	3.48 ± 0.08 ^ab^	3.21 ± 0.12 ^b^	0.008
Pelvis width (cm)	5.76 ± 0.09 ^a^	5.07 ± 0.11 ^b^	5.01 ± 0.13 ^b^	4.76 ± 0.20 ^b^	0.000
Half-diving depth (cm)	33.50 ± 0.57 ^a^	28.24 ± 0.54 ^b^	26.74 ± 0.71 ^bc^	25.19 ± 0.91 ^c^	0.000

Note: The values used for one-way ANOVA analysis are presented as the means ± standard error. The number of geese in the 0 (control), 0.5, 1.0, and 2.0 mg/kg T-2 toxin groups was 12, 9, 10, and 7, respectively. ^a,b,c^ Means within a row with no common superscript differ significantly (*p* < 0.05).

**Table 4 animals-14-02281-t004:** Effects of T-2 toxin on tibial development and bone quality in Yangzhou goslings.

Item	0 mg/kg	0.5 mg/kg	1.0 mg/kg	2.0 mg/kg	*p*-Value
Weight (g)	8.96 ± 0.49 ^a^	7.73 ± 0.29 ^ab^	7.02 ± 0.46 ^bc^	5.49 ± 0.45 ^c^	0.000
Volume (mL)	6.63 ± 0.49 ^a^	5.54 ± 0.27 ^ab^	5.28 ± 0.33 ^ab^	3.93 ± 0.41 ^b^	0.001
Length (mm)	99.25 ± 2.34 ^a^	90.89 ± 1.25 ^ab^	91.03 ± 1.73 ^ab^	84.44 ± 2.39 ^b^	0.000
Width (mm)	6.06 ± 0.12 ^a^	5.59 ± 0.11 ^ab^	5.38 ± 0.15 ^bc^	4.81 ± 0.18 ^c^	0.000
Circumference (cm)	2.26 ± 0.07 ^a^	2.06 ± 0.03 ^ab^	2.00 ± 0.04 ^bc^	1.81 ± 0.06 ^c^	0.000
Breaking strength (N)	120.55 ± 3.96 ^a^	105.08 ± 4.21 ^ab^	91.55 ± 3.47 ^b^	75.37 ± 2.46 ^c^	0.000
Skim weight (g)	2.93 ± 0.14 ^a^	2.43 ± 0.09 ^ab^	2.28 ± 0.15 ^bc^	1.71 ± 0.14 ^c^	0.000
Density (g/cm^3^)	0.56 ± 0.02 ^a^	0.49 ± 0.02 ^ab^	0.48 ± 0.02 ^b^	0.48 ± 0.01 ^b^	0.004
Ash (%)	46.36 ± 0.78 ^a^	42.57 ± 0.75 ^b^	43.51 ± 0.88 ^ab^	42.65 ± 0.84 ^b^	0.005
Ca (%)	22.08 ± 0.32 ^a^	20.70 ± 0.49 ^ab^	19.92 ± 0.37 ^b^	19.51 ± 0.46 ^b^	0.000
P (%)	10.84 ± 0.22 ^a^	10.24 ± 0.31 ^ab^	9.43 ± 0.32 ^b^	10.14 ± 0.24 ^ab^	0.006

Note: The values used for one-way ANOVA analysis are presented as the means ± standard error. The number of geese in the 0, 0.5, 1.0, and 2.0 mg/kg T-2 toxin groups was 12, 9, 10, and 7, respectively. ^a,b,c^ Means within a row with no common superscript differ significantly (*p* < 0.05).

## Data Availability

All data generated or analyzed during this study are included in this published paper.

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
