# Peer review of "Tibial Damage Caused by T-2 Toxin in Goslings: Bone Dysplasia, Poor Bone Quality, Hindered Chondrocyte Differentiation, and Imbalanced Bone Metabolism"

_animals, 2024, doi:10.3390/ani14152281_

Round 1

Reviewer 1 Report

Comments and Suggestions for Authors

Review report;

Title: Tibial damage caused by T-2 toxin in goslings: bone dysplasia, poor bone quality, hindered chondrocyte differentiation, and imbalanced bone metabolism.

The study is generally scientifically valuable. It is an original and beautiful work. The level of contribution to science is high. Valuable results have emerged for waterwings. Suggestions in the text should also be taken into consideration.

Study; Tibial damage caused by T-2 toxin in goslings: bone dysplasia, poor bone quality, inhibited chondrocyte differentiation and unbalanced bone metabolism.

Title: The title is compatible with the content.

Simple Summary: It explains the work. Clear results are given. The purpose is stated.

Abstract: The purpose and method of the study are stated. important results are highlighted. The scientific aspect is good.

Keywords: It explains the work. It is enough to promote the article.

Introduction: Adequate and up-to-date literature was used. The purpose of the study is well explained. According to this fiction, the purpose of the study is stated. What is desired to be achieved at the end of the study is stated. However, the correction specified in the text should be taken into account.

Materials and Methods:

- Why was T-2 toxin given by oral gavage? Wasn't it possible to mix it with feed and water? The answer to the question should be given in the method.

- Analyzes need to be supported by literature.

- Literature is not provided for most analyzes in the methods section. This situation must be corrected.

2.3. Tibial Phenotypic Measurements and Imaging Analysis - Literature should be added. From whom was this method derived?

2.2. Body Measurements - - Literature should be added. From whom was this method derived?

2.4. Tibial Bone-breaking Strength Determination - - Literature should be added. From whom was this method derived?

2.6. Tibial Bone Composition Determination - - Literature should be added. From whom was this method derived?

2.7. Tibial Stainings by Hematoxylin-eosin (HE), Alcian Blue (AB), Safranin O-Fast Green (SF), and Tartrate-Resistant Acid Phosphatase (TRAP) - - Literature should be added. From whom was this method derived?

2.9. Serum Biochemical Tests - - Literature should be added. From whom was this method derived?

Results: Explanatory and realistic data were used in the study. The pictures were beautiful and descriptive. The results are scientifically valuable.

Tables and graphs are understandable. The pictures were good for understanding the article. Picture quality can be improved. This will be important to readers.

Discussion: Literature support was received. Discussion was made according to the data in the results. Literature support is good.

The results and discussion are consistent. It is self-explanatory.

Conclusions: The conclusion section was written according to the study data. It summarizes the discussion.

Owerall:

The study is generally scientifically valuable. It is an original and beautiful work. The level of contribution to science is high. Valuable results have emerged for Waterfowl.

References are appropriate and sufficient for the content of the study. Current literature was used. The method section should be updated by adding literature.

Development is very important, especially in water birds. Determination of toxic effects has made an important scientific contribution.

Author Response

Reviewer 1#

Introduction

Q1: It should start as a separate paragraph.

R1: Thank you for your reminder, and we have done as requested (L75-76).

Materials and Methods

Q2: - Why was T-2 toxin given by oral gavage? Wasn't it possible to mix it with feed and water? The answer to the question should be given in the method.

R2: Thank you for your advice, and we have added relevant information to the manuscript (L143-146).

To reduce the difficulty in uneven stirring, toxin waste, individual feed intake calculation, and individual injury phenotype differences by T-2 toxin supplemented in feed, we fed goslings with T-2 toxin by oral gavage in this study.

T-2 toxin is soluble in ethanol, but not directly soluble in water. T-2 toxin is usually first dissolved in ethanol, then dissolved in water, sprayed into feed, and volatilized for use. We previously found that T-2 toxin supplemented in feed inhibited feed intake and growth and development in goslings. However, the supplement of T-2 toxin in feed causes uneven stirring, toxin waste, individual feed intake calculation, and individual injury phenotype differences. Therefore, we used oral gavage to explore the effect of T-2 toxin on tibial growth and development.

Q3: - Analyzes need to be supported by literature.

- Literature is not provided for most analyzes in the methods section. This situation must be corrected.

R3: Thank you for your suggestion. We have added the relevant references in the manuscript according to your guidance.

Q4: 2.2. Body Measurements - - Literature should be added. From whom was this method derived?

R4: Thank you for your help. we have supplemented the relevant reference [32] in the manuscript (L185).

Q5: 2.3. Tibial Phenotypic Measurements and Imaging Analysis - Literature should be added. From whom was this method derived?

R5: Thank you. we have provided the relevant references [34-36] in the manuscript (L201).

Q6: 2.4. Tibial Bone-breaking Strength Determination - - Literature should be added. From whom was this method derived?

R6: Thank you for your comment, and we have added the relevant references [34,37] in the manuscript (L211).

Q7: 2.6. Tibial Bone Composition Determination - - Literature should be added. From whom was this method derived?

R7: We are grateful for your review and have added the relevant reference [17] in the manuscript (L225,230).

Q8: 2.7. Tibial Stainings by Hematoxylin-eosin (HE), Alcian Blue (AB), Safranin O-Fast Green (SF), and Tartrate-Resistant Acid Phosphatase (TRAP) - - Literature should be added. From whom was this method derived?

R8: We appreciate you and have supplemented the relevant references [38,39] in the manuscript (L235).

Q9: 2.9. Serum Biochemical Tests - - Literature should be added. From whom was this method derived?

R9: Thank you for your help, and we have provided the relevant reference [8] in the manuscript (L262).

Reviewer 2 Report

Comments and Suggestions for Authors

Title: Tibial damage caused by T-2 toxin in goslings: bone dysplasia, poor bone quality, hindered chondrocyte differentiation, and imbalanced bone metabolism. 

Male Yangzhou goslings were randomly divided into four groups and daily gavaged with T-2 toxin at concentrations of 0, 0.5, 1.0, and 2.0 mg/kg body weight for 21 days, respectively. T-2 toxin significantly inhibited tibial growth and development, accompanied by decreased bone geometry parameters and properties, hindered chondrocyte differentiation, and imbalanced bone metabolism. These findings reveal tibial damage caused by T-2 toxin in goslings, providing scientific information for understanding and alleviating osteotoxicity of T-2 toxin in poultry production. The topic is interesting however there are few points needed to be corrected and revised. 

Add the experimental design to the abstract.

Add P values to the abstract.

Revise the conclusion in the abstract.

L56: commonly? I don't agree with that.

L57: explain why? more specific details are required.

The introduction needed to be more organized, the paragraphs are not clear.

L135-136: rewrite.

L136-137: more information is required about T-2 toxin

Table 1: the requirement was based on what reference? NRC, 1994 or breeds guide? give more details.

L159: write the specific age of goslings.

L162: serum or plasma? more details are required.

L172-184; 187-192; 195-200, 202-216; 245-248: add references and more specific details.

L245-248: for what purpose.

Revise the discussion part, reorganize. Add more links between your findings and the literature.

The conclusion: be more specific based on your own results, what dose of T-2 is harmful? revise and rewrite. 

Comments on the Quality of English Language

Minor corrections in grammar and writing style are required. 

Author Response

Reviewer 2#

Title: Tibial damage caused by T-2 toxin in goslings: bone dysplasia, poor bone quality, hindered chondrocyte differentiation, and imbalanced bone metabolism. 

Male Yangzhou goslings were randomly divided into four groups and daily gavaged with T-2 toxin at concentrations of 0, 0.5, 1.0, and 2.0 mg/kg body weight for 21 days, respectively. T-2 toxin significantly inhibited tibial growth and development, accompanied by decreased bone geometry parameters and properties, hindered chondrocyte differentiation, and imbalanced bone metabolism. These findings reveal tibial damage caused by T-2 toxin in goslings, providing scientific information for understanding and alleviating osteotoxicity of T-2 toxin in poultry production. The topic is interesting however there are a few points needed to be corrected and revised. 

Response: Thank you for your positive evaluation and constructive critiques. We have revised the manuscript point-to-point following your suggestions. The revised portions are marked in red highlighting. We hope this revised version is acceptable for publication in the animals journal.

Add the experimental design to the abstract.

Response: We apologize for not highlighting the experimental design in the abstract and have supplemented the relevant information in the manuscript (L31-39).

Add P values to the abstract.

Response: Thank you for your suggestion, and we have added P values in the abstract (L41-52).

Revise the conclusion in the abstract.

Response: Thank you for your advice, and we have revised the conclusion (L52-60). We supplemented the detailed information on the T-2 toxin dose effect and tibia damage. Significant adverse effects of T-2 toxin on tibial development, chondrocyte differentiation, and bone metabolism in the high-dose group (2.0 mg/kg) were revealed compared to the control and low-dose groups (0.5 and 1.0 mg/kg).

L56: commonly? I don't agree with that.

Response: Thank you for your reminder. We have deleted this word to reduce the misunderstanding of the high contamination rate of T-2 toxin in animal feed (L64-65).

L57: explain why? more specific details are required.

Response: Thank you for your reminder. We have provided more specific details of the high sensitivity of goslings to T-2 toxin (L66-68).

The introduction needed to be more organized, the paragraphs are not clear.

Response: Thank you for your constructive comments, we have revised the layout of the introduction to make it clearer and more logical. Here, we follow the logic from the concern of industrial problems to the tibial phenotype of pathological damage to the analysis of the damage mechanisms. Finally, we present the object, purpose, and significance of our study. If we still have some problems, we request for your further guidance.

       In the first paragraph of the introduction, we show that T-2 toxin is a threat to normal growth and development in goslings and that bone is a target of damage (L64-74). In the second paragraph, we reveal that changes in tibial phenotype and characteristics (growth and developmental status and bone quality) under toxin exposure are important phenotypic data for revealing the osteotoxicity of T-2 toxin (L75-95). In the third paragraph, investigating the growth plate tibia contributes to unraveling the intrinsic causes of tibial longitudinal growth retardation under T-2 toxin exposure (L96-113). In the fourth paragraph, the study of bone metabolism can help reveal the underlying mechanism of changes in tibial bone quality and health in response to T-2 toxin exposure (L114-124). In the last paragraph, we present the object, purpose and significance of this study (L125-131).

L135-136: rewrite.

Response: Thanks for your comment. We have done as requested (L148-150).

L136-137: more information is required about T-2 toxin

Response: Thanks for your advice. We have supplemented the physicochemical properties of T-2 toxin (L150-151). If we still cause some misunderstandings, please help us to supplement it in detail.

Table 1: the requirement was based on what reference? NRC, 1994 or breeds guide? give more details.

Response: The diet in Table 1 was based on the breed guide of Yangzhou goose in China (GB/T 36784-2018). Nutritional levels have been optimized to prevent gout in goslings, such as decreased crude protein and Ca. We have provided the relevant information in the manuscript (L142-143).

L159: write the specific age of goslings.

Response: We are sorry for not mentioning the age of the goslings, and have supplemented the relevant information (21-day-old; L169).

L162: serum or plasma? more details are required.

Response: We collected serum samples for serum biochemical tests and enzyme-linked immunosorbent assay (ELISA). We have supplemented the relevant information (L172).

L172-184; 187-192; 195-200, 202-216; 245-248: add references and more specific details.

Response: We apologize for the insufficient references and some details in the materials and methods. Here, we have provided the relevant information (L182-262).

L245-248: for what purpose.

Response: Goslings receive adequate calcium and phosphorus through their diets to provide raw materials for bone growth and development, such as mineralization and bone metabolic processes. The balance of serum calcium and phosphorus levels is regulated by hormones such as PTH, CT, 1,25-(OH)2-D3. In addition, tibia serves as a storage pool for calcium and phosphorus to maintain stable serum calcium and phosphorus levels. Therefore, elucidating changes in serum calcium and phosphorus levels in response to T-2 toxin exposure is beneficial in determining tibial growth and development and tibial damage. Here, we have determined serum calcium and phosphorus concentrations using serum biochemical tests rather than ELISA, which is a convenient and rapid method.

Revise the discussion part, reorganize. Add more links between your findings and the literature.

Response: Thank you for your constructive comments. Similar to the introduction, we have revised the layout of the discussion to make it clearer and more logical. We follow the logic from the concern of industrial problems to the tibial phenotype of pathological damage to the analysis of the damage mechanisms. In terms of phenotype, T-2 toxin resulted in a decrease in body weight and body size with a dose effect, suggesting bone damage. The tibia, a representative of the long bones, is a target for toxin damage. We further found that tibial development and characteristics were also decreased in a dose manner. To further reveal the mechanism of damage to longitudinal tibial growth, we explored the changes in TGP in terms of phenotype, cellular morphology, and gene expression patterns. In addition, homeostasis of bone metabolism is fundamental for maintaining bone physiologic function and health. We identify that T-2 toxin induces increased osteoblastogenesis and decreased osteoclastogenesis.

In the first paragraph of the discussion, we show that T-2 toxin inhibits weight gain and body development, a valid marker of bone disorders (L407-437). In the second paragraph, we find T-2 toxin induces poor tibial growth and developmental and bone characteristics (L438-463). In the third paragraph, we clarify TGP chondrocyte damage by phenotypic and cytomorphologic observation and analysis, and gene-expression detection. Chondrocytes in TGP were disorganized, indistinctly bordered, and sparse in number under T-2 toxin exposure, accompanied by more vacuoles, excessive apoptosis, necrosis, and poor calcification. (L464-499). In the last paragraph, we identify the inhibition of bone formation of osteoblasts and the enhancement of bone absorption of osteoclasts under T-2 toxin exposure (L500-557). In addition, We emphasize that T-2 toxin exhibits a dose-dependent effect, with the most significant effect at 2.0 mg/kg. If we still have some problems, we request your further guidance in detail.

The conclusion: be more specific based on your own results, what dose of T-2 is harmful? revise and rewrite.

Response: Thank you for your advice, and we have revised the conclusion (L559-569). We supplemented the detailed information relative to the T-2 toxin dose effect and tibia damage. Compared to the control and low-dose groups, the high dose of 2.0 mg/kg of T-2 toxin had a significant negative effects on tibial development, bone quality, chondrocyte differentiation, and the balance of bone metabolism.

Reviewer 3 Report

Comments and Suggestions for Authors

The manuscript by Gu et al., shows the results of a study of T2 toxin on the tibia of growing geese. It is a work whose results can be of great interest for poultry production; Its introduction includes updated bibliography and covers the different aspects that are analyzed in the research. A diagram is presented that allows the experimental design to be clearly interpreted; However, there are serious problems in the presentation of the histological aspects of the paper that are found both in materials and methods and in results and that make the discussion, related to these aspects, not valid.

The authors propose to perform a count of osteoblasts and osteoclasts, but in an organ that is heterogeneous in its structure such as a bone, it is essential to clearly establish which regions of the organ will be studied. It would also be important to clearly establish how many images were taken in each case and what criteria were considered for cell recognition. In the images presented, the cells cannot be recognized; it is necessary to show higher magnifications. The results mention changes such as cellular vacuolization, necrosis and apoptosis, which cannot be recognized in the figures. In addition to including higher magnifications, a bar must be used to interpret the magnification.

Some aspects should be clarified by the authors; For example, because they only took two genes from the BCL-2 family in the analysis, the decrease in the expression of Bcl2, not accompanied by the increase in Bak, is not explained; perhaps the analysis of other proapototic members of the family could clarify this aspect.

Aspects related to the microscopic study must be improved for possible publication in Animals

Author Response

Reviewer 3#

The manuscript by Gu et al., shows the results of a study of T2 toxin on the tibia of growing geese. It is a work whose results can be of great interest for poultry production; Its introduction includes updated bibliography and covers the different aspects that are analyzed in the research. A diagram is presented that allows the experimental design to be clearly interpreted; However, there are serious problems in the presentation of the histological aspects of the paper that are found both in materials and methods and in results and that make the discussion, related to these aspects, not valid.

Response: Thank you for your positive evaluation and constructive critiques of our manuscript entitled “Tibial damage caused by T-2 toxin in goslings: bone dysplasia, poor bone quality, hindered chondrocyte differentiation, and imbalanced bone metabolism” (manuscript ID: animals-3114933). Those comments are valuable and helpful for improving our paper and provide new insights and guidance in further research.

We have revised the manuscript point-to-point following your suggestion, especially the supplemental information related to the histological aspects. The revised portions are marked in red highlighting. We hope this revised version is acceptable for publication in the animals journal.

The authors propose to perform a count of osteoblasts and osteoclasts, but in an organ that is heterogeneous in its structure such as a bone, it is essential to clearly establish which regions of the organ will be studied. It would also be important to clearly establish how many images were taken in each case and what criteria were considered for cell recognition. In the images presented, the cells cannot be recognized; it is necessary to show higher magnifications. The results mention changes such as cellular vacuolization, necrosis and apoptosis, which cannot be recognized in the figures. In addition to including higher magnifications, a bar must be used to interpret the magnification.

Response: We apologize for not clarifying the observation regions, image numbers, and criteria for a count of osteoblasts and osteoclasts. Here, we have added the relevant information and provided Supplementary Figure1-2 of higher magnifications for analysis. Figure 3 and Figure 4 are composed of many small graphs, which results in unclear cell morphology and scale bar. Even if we use the enlarged figure, it still causes misunderstandings. Therefore, we use the enlarged figure as Supplementary Figures to provide detailed information about the cell morphology. If there are still some problems, we look forward to your further guidance.

For the count analysis of osteoblasts and osteoclasts in Figure 4, we chose the mineralized region of TGP for our observations. In the mineralized area, late hypertrophic chondrocytes and newly generated bone trabeculae are obvious markers. Osteoblasts are usually polygonal or cuboidal in shape, arranged in a single layer on the surface of the newly formed bone matrix and involved in the formation of new bone. It has a round or oval nucleus, which is usually positioned on one side of the cell. The surface of the cell is characterized by numerous cellular protrusions, which contribute to intercellular material exchange and signaling. Osteoclasts are large multinucleated cells (n ≥ 3) that usually show intense red or purple coloring in TRAP staining. Osteoblasts and osteoclasts were counted in five randomly selected fields of HE and TRAP staining respectively with a scale bar of 100 μm, and the mean values were used as a representative value for the individual sample within the group for subsequent significance analysis (one-way ANOVA analysis). In addition, the representative images of higher magnifications of osteoblasts and osteoclasts with a scale bar of 20 μm were shown in Supplementary Figure 2.

For the morphologic analysis of TGP chondral damage in Figure 3, we displayed different regions of continuity throughout the TGP in a single figure, presenting morphological changes throughout the TGP under T-2 toxin exposure. HE staining was used as the basic cytomorphological observation and analysis with a scale bar of 1000 μm, and AB and SF specificity stains were used to mark and differentiate chondrocytes. However, it obstructed the identification of cellular damage (cellular vacuolization, necrosis, and apoptosis). Cellular vacuolization or cell loss is mainly manifested by multiple vacuoles of varying sizes within the cytoplasm, which usually appears as unstained transparent areas under HE staining. In addition, morphological features of apoptosis include reduced cell size, nuclear chromatin condensation and nuclear fragmentation, increased cytoplasmic eosinophilia, and formation of apoptotic vesicles. Cellular necrosis is characterized by cellular swelling, nuclear consolidation, nuclear fragmentation and nucleolysis, increased cytoplasmic eosinophilia, rupture of cell membranes, disruption of tissue architecture, and infiltration of inflammatory cells. Here, we have provided representative images of higher magnifications of TGP chondral damage with a scale bar of 20 μm in HE staining (Supplementary Figure 1).

Some aspects should be clarified by the authors; For example, because they only took two genes from the BCL-2 family in the analysis, the decrease in the expression of Bcl2, not accompanied by the increase in Bak, is not explained; perhaps the analysis of other proapototic members of the family could clarify this aspect.

Response: Thank you for your suggestion. T-2 toxin-induced apoptosis in TGP chondrocytes was observed by the morphology analysis under various stainings. Moreover, we validated it through changes in the expression of apoptosis-related genes. The ratio of BAX/BCL2 is a key indicator for evaluating the status of apoptosis. Increased BAX expression promotes apoptosis, whereas increased BCL2 expression inhibits apoptosis. However, there is no BAX gene data in the NCBI genebank of goose. We therefore determined the gene expression changes of BAK1. The non-significant expression of the BAK1 gene suggests the involvement of other pro-apoptotic genes, warranting future research. Interestingly, T-2 toxin induced a significant decrease in BCL2 expression in TGP, suggesting a decrease in anti-apoptotic capacity. In addition, gene expression of CASP9 (caspase-9) and CASP3 (caspase-3), key enzymes in the apoptotic pathway, which are involved in apoptotic signaling and execution phases respectively, was significantly increased. Therefore, the expression of these apoptosis-related genes can indicate that T-2 toxin induces excessive apoptosis in chondrocytes, while the study of gene expressions of the BCL2 family is limited.

Apoptosis is a complex physiological process that is regulated by multiple genes to maintain normal development and health. As you mentioned, clarifying the expression changes of more apoptosis-related genes could help to better explore the osteotoxicity of T-2 toxin. In future studies, we will aim to determine the expression of many other relevant key genes, such as BOK, BNIPs, and BID. However, this requires extensive investigation or could be identified using RNA-seq. We have also added a description in the Discussion (L491-495). If there are still some problems, we look forward to your further guidance.

Aspects related to the microscopic study must be improved for possible publication in Animals

Response: Thanks for the reminder and constructive comments. As you requested, we have optimized the microscopic analysis with the relevant attachments (Supplementary Figure 1-2). We hope that it will meet your requirements and be able to publish in the animal journal. If there are still some problems, we look forward to your further guidance.

Reviewer 4 Report

Comments and Suggestions for Authors

Title: Tibial damage caused by T-2 toxin in goslings: bone dysplasia, poor

bone quality, hindered chondrocyte differentiation, and imbalanced bone

metabolism

Authors: Wang Gu, Lie Hou, Qiang Bao, Qi Xu *, Guohong Chen *

The authors aim to Therefore, the objective of this study was to investigate the adverse effects of T-2 toxin on bone development, bone quality, chondrocyte differentiation, and bone metabolism in the tibia of Yangzhou goslings. The authors found that  T-2 toxin significantly
inhibited tibial growth and development, accompanied by decreased bone
geometry parameters and properties, hindered chondrocyte differentiation, and
imbalanced bone metabolism. . Even though this work provided interesting results, several concerns need to be made clear.

Please see the comment in the file pdf attached.

Comments on the Quality of English Language

Author Response

Reviewer 4#

Abstract

Q1: Please add research objective.

R1: Thank you for your suggestion. We apologize for not highlighting the objective of the study and have supplemented the relevant information to the manuscript (L31-32).

Q2: Ca and P?

R2: Yes, “calcium and phosphorus” are the full names of “Ca and P”. Thank you for your reminder. To reduce misunderstanding, we use abbreviations according to your guidance (L45).

Q3: This statement should place at section of conclusion and recommendation at the end of manuscript. Present section should report on the key result and not for the outcome of research.

R3: We are grateful for your comment and have deleted the relevant information (L60).

Introduction

Q4: Is this statement was evaluated in present work? If no, recommend to remove.

R4: Thank you for your advice. To reduce misunderstanding, we have deleted the relevant information (L131).

Materials and Methods

Q5: Add DM content

R5: Thank you for your recommendation, and we have added DM content (87.49%) in Table 1.

Results

Q6: Please insert the p-values of each observation into another column. Check for all data tables.

R6: Thank you for your suggestion, and we have added the p-values in all data tables (Table 3, Table 4).

Discussion

Q7: Provide biological mechanism how these factor can be affect BW?

R7: Thank you for your help. we have supplemented the biological mechanism of BW loss under T-2 exposure (L415-424).

Q8: Too long sentence, please separate into 2 sentences.

R8: Thanks for your suggestion. We have simplified the long sentence into two shorter sentences, preserving the original meaning while making the expression clearer and easier to understand. (L495-499).

Conclusions

Q9: When? Low or high?

R9: Thanks for your comment. In this study, we explored the effect of T-2 toxin on tibial growth and development, morphological structure, and characteristics in goslings after 21 days of oral gavage. Results showed that bone geometry parameters including the tibial weight, volume, length, width, and circumference were significantly lower under 2.0 mg/kg T-2 toxin exposure. In addition, the bone properties including bone-breaking strength, density, and inorganic composition exhibited a similar tendency. Therefore, we summarized it as “decreased bone geometry parameters and properties” in the conclusion. To reduce the misunderstanding, we have optimized the conclusion.

Round 2

Reviewer 2 Report

Comments and Suggestions for Authors

Thank you for providing revised manuscript. 

Reviewer 3 Report

Comments and Suggestions for Authors

In the present form the manuscript can be accepted

Reviewer 4 Report

Comments and Suggestions for Authors

The authors have made revisions in accordance with my earlier comments, and no new suggestions have been made.